# PercepLIE: A New Path to Perceptual Low-Light Image Enhancement

Cong Wang
Shenzhen Campus of Sun Yat-sen University & The Hong
Kong Polytechnic University
Shenzhen & Hong Kong, China
supercong94@gmail.com

Chengjin Yu
Anhui University
Hefei, China
23073@ahu.edu.cn

Jie Mu
Dongbei University of Finance and Economics
Dalian, China
jiemu@dufe.edu.cn

Wei Wang*
Shenzhen Campus of Sun Yat-sen University
Shenzhen, China
wangwei29@mail.sysu.edu.cn

## Abstract

While current CNN-based low-light image enhancement (LIE) approaches have achieved significant progress, they often fail to generate better perceptual quality which requires restoring better details and more natural colors. To address these problems, we set a new path, called PercepLIE, by presenting the VQGAN with Multi-luminance Detail Compensation (MDC) and Global Color Adjustment (GCA). Specifically, observed that latent light features of the low-light images are quite different from those captured in normal light, we utilize VQGAN to explore the latent light representation of normal-light images to help the estimation of the low-light and normal-light mapping. Furthermore, we employ Gamma correction with varying Gamma values on the gradient to create multi-luminance details, forming the basis for our MDC module to facilitate better detail estimation. To optimize the colors of low-light input images, we introduce a simple yet effective GCA module that is based on spatially-varying representation between the estimated normal-light images in this module and low-light inputs. By combining the VQGAN with MDC and GCA within a stage-wise training mechanism, our method generates images with finer details and natural colors and achieves favorable performance on both synthetic and real-world datasets in terms of perceptual quality metrics including NIQE, PI, and LPIPS. The source codes will be made available at https://github.com/supersupercong/PercepLIE.

## CCS Concepts

• **Computing methodologies → Computational photography**.

*Wei Wang is the Corresponding author.

## Keywords

Low-Light Image Enhancement, Latent Light Representation, Multi-luminance Detail Compensation, Global Color Adjustment

**ACM Reference Format:**
Cong Wang, Chengjin Yu, Jie Mu, and Wei Wang. 2024. PercepLIE: A New Path to Perceptual Low-Light Image Enhancement. In *Proceedings of the 32nd ACM International Conference on Multimedia (MM '24), October 28–November 1, 2024, Melbourne, VIC, Australia.* ACM, New York, NY, USA, 10 pages. https://doi.org/10.1145/3664647.3681399

## 1 Introduction

Low-light image enhancement (LIE) aims to recover a normal-light image from the observed low-light one. LIE attracts lots of attention due to its significant applications in computer vision, video surveillance, and multimedia. Early works on LIE are based on statistical observations, e.g., histogram equalization [3, 48, 75], gamma correction [13, 50], and Retinex model [7, 35]. However, statistical observations do not sufficiently model the inherent proprieties of clear natural images and may lead to unnatural results.

Recent LIE approaches [9, 36, 43, 63, 66, 82] mainly rely on deep convolutional neural networks (CNNs) [11, 12, 31], by designing various networks with end-to-end training to directly obtain the restored results from the low-light input images. Although these CNN-based approaches achieve decent performance compared to conventional methods based on statistical observations, there remain critical challenges that need to be addressed.

**Challenges.** First, the latent light representation of the low-light images is quite different from those captured in normal light, as shown in Figure 1. Directly applying a deep model to low-light images will not effectively explore useful features for image enhancement. Thus, it is necessary to resort to the deep light representation extracted from normal light to guide the deep model for better restoration. Second, since recovering details is essential for image restoration, most existing methods [9, 15, 39, 63, 67, 82] are not effective for image details estimation. Although some approaches [47, 85] try to use structures to guide the estimation process of clear images, the adopted structures are usually decimated when applying convolution operations recurrently, thus may be not effective for better detail estimation. Third, LIE aims to generate images with not only fine structural details but also natural colors and better perceptual quality. Existing approaches do not effectively

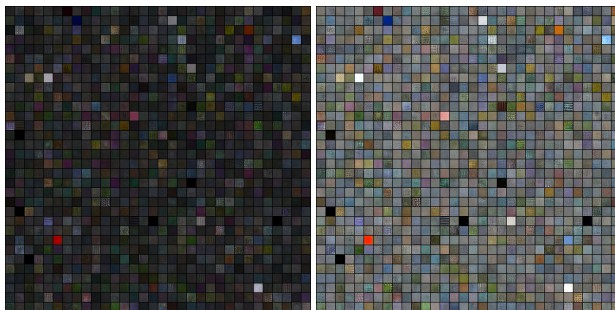

(a) Codebook of low light  (b) Codebook of normal light

**Figure 1: Latent light representation (i.e., the codebook) learned by VQGAN based on low-light images and normal-light images, respectively. We empirically find that the latent light representation of low-light images is quite different from those of images captured in normal light. Hence, enforcing normal-light codebook priors to guide the deep models will help better enhance.**

model various illumination distributions of low light, which usually leads to results with unnatural colors and lower perceptual quality. Thus, it is imperative to develop methods that can recover images with better structural details while having natural colors and higher perceptual quality.

**Solutions.** To solve the above challenges, we in this paper propose the **PercepLIE**, which is a VQGAN-based model with Multi-luminance Detail Compensation (MDC) and Global Color Adjustment (GCA) for perceptual LIE. As noted that the latent features of the low-light images are quite different from those captured in normal light, as shown in Figure 1. VQGAN [5] is used to explore the latent light representations of normal-light images via learning the normal-light codebook priors to guide the deep models to enhance images by reducing the estimation error of the low-light and normal-light mapping (Section 4.1). To generate the results with finer details, we apply Gamma correction with different gamma values to generate multi-luminance details based on image gradient, as shown in Figure 2, and develop an effective MDC scheme to compensate for the details for better detail estimation (Section 4.2). However, due to the influence of different illumination conditions, the restored images usually contain unnatural colors. We thus develop a simple yet effective GCA approach, which is built on the spatially-varying representation between the estimated images and low-light ones, to adjust the restored results toward more natural colors (Section 4.3). By training the VQGAN with MDC and GCA stage by stage, our PercepLIE can generate high-quality and realistic-looking images.

**Evaluation Criteria.** The objective of LIE is to recover natural-looking and perceptually pleasing results from low-light inputs. However, we note that widely used distortion metrics such as Peak-Signal-to-Noise-Ratio (PSNR) [14] and Structural SIMilarity (SSIM) [64] are inadequate for measuring these properties [1, 32]. Notice that perceptual quality metrics such as Natural Image Quality Evaluator (**NIQE**) [44] and Perceptual Indexes (**PI**) [41] can estimate the perceptual quality and naturalness of the restoration result [9, 79]. Additionally, the Learned Perceptual Image Patch Similarity (**LPIPS**) [78] has been shown to closely match human

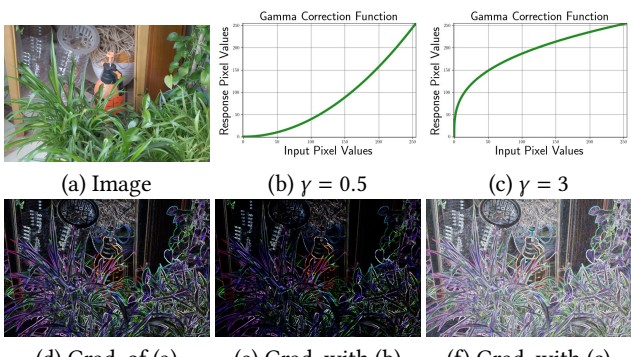

(a) Image  (b) $\gamma = 0.5$  (c) $\gamma = 3$

(d) Grad. of (a)  (e) Grad. with (b)  (f) Grad. with (c)

**Figure 2: Multi-luminance Details. We empirically find that the original gradient (d) preserves the details of the original image. Gamma correction ($y = x^\gamma$) with smaller $\gamma$ ($<1$) can produce the main structure while larger $\gamma$ ($>1$) is able to find more details. The luminance gradient forms the multiple details that will be useful for better detail compensation in image restoration.**

perception, making it an appropriate choice to evaluate the perceptual similarity between two images. Therefore, to assess restoration quality, we employ **NIQE**, **PI**, and **LPIPS** as the evaluation metrics in this study.

Our contributions are summarized as follows:

- We propose a VQGAN-based LIE model to generate high-quality images by exploring the latent light representations of normal-light images to facilitate the estimation of the mapping between the low-light and normal-light images.
- We propose a multi-luminance detail compensation module (MDC) by estimating multi-luminance details to adaptively compensate for details of the generated images for high-quality image enhancement.
- We propose a global color adjustment module (GCA) by introducing the spatially-varying representation between the estimated normal-light images in this module and low-light ones to adjust the results toward more natural colors.
- We formulate the VQGAN with MDC and GCA in a new formulation for perceptual LIE, and show our method achieves favorable performance on both synthetic and real-world datasets in terms of various perceptual quality metrics.

## 2 Related Work

### 2.1 Low-Light Image Enhancement

Early research on LIE mainly utilizes histogram equalization [3, 48, 75], sparse representation model [6], and Retinex theory [73]. Benefiting from the powerful learning mapping ability of CNNs for visual problems [4, 16–21, 23–29, 53, 54, 56–59, 68–70, 74, 84], recent research on LIE mainly focuses on designing varieties of deep neural networks. Inspired by Retinex theory [46], some Retinex-based networks are developed [39, 66, 82]. To recover finer structures of enhanced images, a series of works have been proposed [47, 55, 71, 85]. By formulating the LIE task as a deep curve estimation problem [9] or illumination adjustment problem [43], some unsupervised algorithms are proposed [9, 25, 36, 43]. Despite these efforts, these

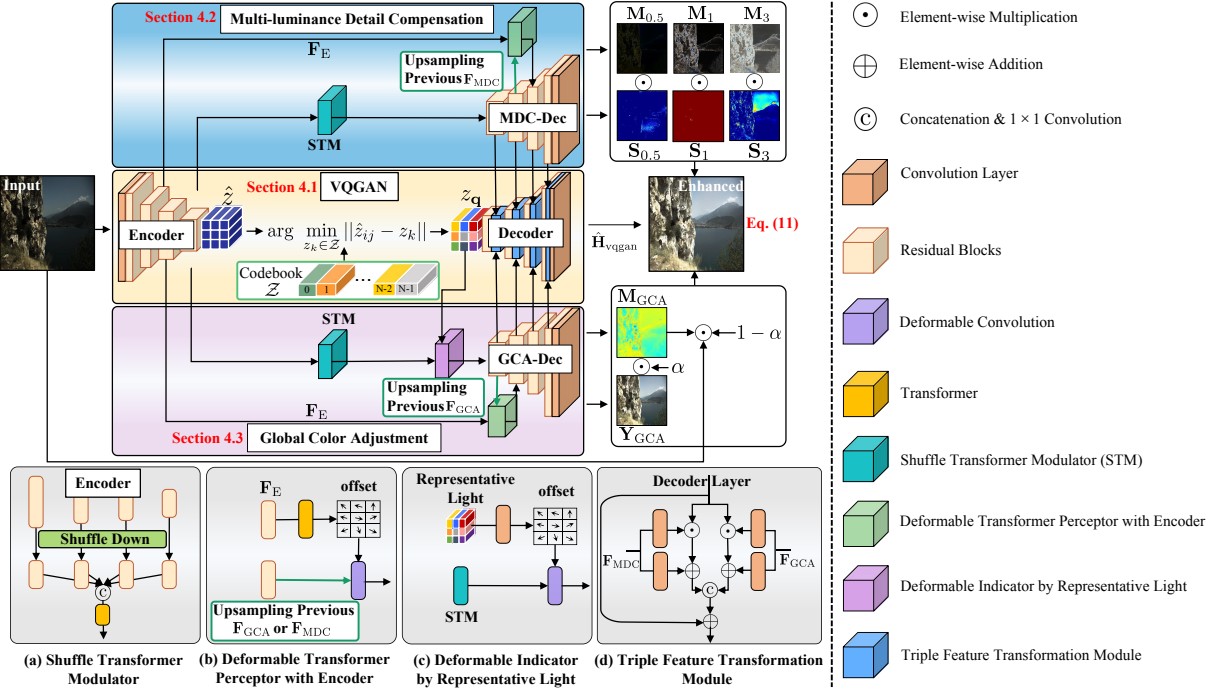

**Figure 3: Paradigm of our PercepLIE. The PercepLIE is based on the VQGAN with multi-luminance detail compensation (MDC) and global color adjustment (GCA). The VQGAN explores the latent light representation of normal-light images to help better estimation of the low-light and normal-light mapping (Section 4.1). The MDC estimates multi-luminance details for better compensating details (Section 4.2). The GCA controls the generation results by building the spatially-varying representation between the estimated images in GCA and low-light ones to control the results for better colors and naturalness (Section 4.3).**

approaches may not generate results with better perceptual qualities and natural colors since they either do not explore the latent normal-light priors or cannot adjust the results toward natural colors.

## 2.2 VQGAN and Its Applications to Restoration

Esser et al. [5] propose to train the vector-quantized codebook [51] with the adversarial objective to achieve higher perceptual quality. Inspired by its ability in generating high-quality images, some recent works exploit VQGAN for image restoration [2, 8, 52, 65, 83]. To our knowledge, there has been no effort to exploit VQGAN for LIE. To better explore LIE, we first find that the latent feature space of low-light images is quite different from those of images captured in normal light. Hence, the VQGAN can be utilized to learn the codebook to guide the deep model. With the VQFAN, we propose multi-luminance detail compensation to compensate for details and global color adjustment to dynamically adjust the results for better natural colors according to different low-light observations.

## 3 Motivation

To better motivate our method, we first revisit the learning-based formulation for LIE in Section 3.1, then introduce our new path to problem formulation for perceptual LIE in Section 3.2.

## 3.1 Learning-based Formulation Revisited

We first revisit the learning-based problem formulation of LIE. Most existing approaches [22, 72] directly map the low-light images $\mathbf{L}$ to

enhance ones $\hat{\mathbf{H}}$ via a deep model $\mathcal{N}$:

$$\hat{\mathbf{H}} = \mathcal{N}(\mathbf{L}) \tag{1}$$

We note that (1) directly maps low-light inputs to estimated normal-light ones without considering the details and colors of generated results, which may lead to more unnatural results and worse perceptual quality as high-quality image restoration requires recovering results with both finer details and natural colors. However, the details are usually decimated when applying convolution operations recurrently and the colors of restored results may not be promising when handling various low-light scenarios with different illuminations. Hence, the formulation in (1) without considering both details and colors may not be able to recover more natural results.

## 3.2 A New Path to Problem Formulation

To solve these challenges in Section 3.1, we explore a new path to perceptual LIE by taking image content, details, and colors into consideration within a new formulation:

$$\hat{\mathbf{H}} = \mathcal{N}_{\text{content}}(\mathbf{L}) + \mathcal{N}_{\text{detail}}(\mathbf{L}) + \mathcal{N}_{\text{color}}(\mathbf{L}) \tag{2}$$

where $\mathcal{N}_{\text{content}}$ is used to restore image content of normal-light; $\mathcal{N}_{\text{detail}}$ is used to compensate for details; $\mathcal{N}_{\text{color}}$ is used to adjust the colors of the enhanced results.

To achieve the above goals, we propose the **PercepLIE**, which consists of 1) VQGAN for exploring latent light representation, which is used to better help generate image content (Section 4.1); 2)

a multi-luminance detail compensation module, which can better compensate for details (Section 4.2); 3) a global color adjustment module, which can help better adjust the colors (Section 4.3). Figure 3 illustrates the framework of our PercepLIE.

## 4 PercepLIE

As introduced in Section 1, our method mainly contains a VQGAN for exploring the latent light representation of normal-light images via learning the normal-light codebook, a multi-luminance detail compensation module to compensate details, and a globally controllable adjustment module to control the generated results with natural colors and perceptual quality. Figure 3 illustrates our method.

### 4.1 Latent Light Representations

We note that the latent light representations, i.e., codebook, of the low-light images are quite different from those captured in normal light, as shown in Figure 1. Hence, enforcing this characteristic to guide the deep models to enhance images will be helpful for LIE. To that end, we employ the VQGAN [5] to learn the Vector-Quantized (VQ) codebook. The VQGAN consists of an encoder $E$, a decoder $D$, and a codebook $\mathcal{Z} = \{z_k\}_{k=1}^{N}$ with $N$ discrete codes. Given a normal-light image $\mathbf{H} \in \mathbb{R}^{H \times W \times 3}$, we first use an encoder $E$ to extract feature $\hat{z} = E(\mathbf{H}) \in \mathbb{R}^{h \times w \times n_z}$ from $\mathbf{H}$, where $n_z$ is the feature dimension. We then obtain the vector-quantized representation $z_\mathbf{q}$ by applying an element-wise quantization $\mathbf{q}(\cdot)$ of each spatial code $\hat{z}_{ij} \in \mathbb{R}^{n_z}$ to its closest codebook entry $z_k$:

$$z_\mathbf{q} = \mathbf{q}(\hat{z}) := \left( \arg \min_{z_k \in \mathcal{Z}} ||\hat{z}_{ij} - z_k|| \right) \in \mathbb{R}^{h \times w \times n_z}. \quad (3)$$

The decoder $D$ uses the quantized representation $z_\mathbf{q}$ to reconstruct image $\hat{\mathbf{H}}$:

$$\hat{\mathbf{H}} = D(z_\mathbf{q}) = D(\mathbf{q}(E(\mathbf{H}))). \quad (4)$$

We train the VQGAN and codebook in an end-to-end manner and use the following loss function to constrain the VQGAN and codebook:

$$\mathcal{L}(E, D, \mathcal{Z}) = \mathcal{L}_\text{vq} + \lambda_\text{per} \mathcal{L}_\text{per} + \lambda_\text{adv} \mathcal{L}_\text{adv}, \quad (5)$$

where $\lambda_\text{per}$ and $\lambda_\text{adv}$ are weight parameter. $\mathcal{L}_\text{vq}$, $\mathcal{L}_\text{per}$, and $\mathcal{L}_\text{adv}$ are defined as:

$$\mathcal{L}_\text{vq} = ||\mathbf{H} - \hat{\mathbf{H}}||_1 + ||\text{sg}[E(\mathbf{H})] - z_\mathbf{q}||_2^2 + \beta ||\text{sg}[z_\mathbf{q}] - E(\mathbf{H})||_2^2, \quad (6a)$$

$$\mathcal{L}_\text{per} = ||\Phi(\hat{\mathbf{H}}) - \Phi(\mathbf{H})||_2^2, \quad (6b)$$

$$\mathcal{L}_\text{adv} = \mathbb{E}_\mathbf{H}[\log \mathcal{D}(\mathbf{H})] + \mathbb{E}_{\hat{\mathbf{H}}}[1 - \log \mathcal{D}(\hat{\mathbf{H}})]. \quad (6c)$$

Here $||\mathbf{H} - \hat{\mathbf{H}}||_1$ is the reconstruction loss. sg[·] denotes stop gradient operation. $\Phi(\cdot)$ denotes the feature extractor of VGG19 [49]. $\beta$ is a weight parameter, which is empirically set to be 0.25 in all experiments. The codebook is updated by $||\text{sg}[E(\mathbf{H})] - z_\mathbf{q}||_2^2 + \beta ||\text{sg}[z_\mathbf{q}] - E(\mathbf{H})||_2^2$, which is the commitment loss [51].

With the learned codebook of normal light which better represents the latent light feature, we can use it to guide the learning of the model in the next stages as we observe that the codebook of low light is different from those of normal light (Figure 1). Hence, enforcing normal-light codebook priors to guide the deep models will help better enhancement.

## 4.2 Multi-luminance Detail Compensation

We note that applying the Gamma correction with different Gamma $\gamma$ on image gradients leads to multi-luminance results. These multi-luminance results contain different image details, as shown in Figure 2. The details from the multi-luminance outputs by the Gamma correction are complementary, which can provide finer details to guide the deep models for better detail estimation.

To this end, we propose a Multi-luminance Detail Compensation (MDC) (see Figure 3) under the guidance of the multi-luminance outputs by the Gamma correction to better estimate details. Specifically, given a low-light input $\mathbf{L} \in \mathbb{R}^{H \times W \times 3}$, MDC first receives the features from the encoder of VQGAN. Then the MDC utilizes the shuffle Transformer fusion, as shown in Figure 3(a), to aggregate features of different levels in the encoder of VQGAN. And then, the aggregated features are input to the MDC decoder (MDC-Dec). To better exploit and explore useful features in the encoder of VQGAN, we propose a Deformable Transformer Interaction with Encoder (DTP-Enc) module to improve the feature representation of the MDC decoder. Finally, MDC outputs estimated multiple luminance details $\mathbf{S} \in \mathbb{R}^{H \times W \times 3}$ and corresponding weight maps $\mathbf{M} \in \mathbb{R}^{H \times W \times 1}$:

$$\{\mathbf{M}_{0.5}, \mathbf{S}_{0.5}; \mathbf{M}_1, \mathbf{S}_1; \mathbf{M}_3, \mathbf{S}_3\} = MDC(\mathbf{F}_E), \quad (7)$$

where $\mathbf{F}_E$ denotes the features from the encoder in the VQGAN. As ground truth maps are not available, we do not impose any constraint on the weight maps. We use the learned details with corresponding weight maps to adaptively compensate the output of VQGAN $\hat{\mathbf{H}}_\text{vqgan}$ by:

$$\hat{\mathbf{H}} = \hat{\mathbf{H}}_\text{vqgan} + \underbrace{\mathbf{M}_{0.5} \cdot \mathbf{S}_{0.5} + \mathbf{M}_1 \cdot \mathbf{S}_1 + \mathbf{M}_3 \cdot \mathbf{S}_3}_{\text{multi-luminance detail compensation}}. \quad (8)$$

At this stage, parameters in the encoder of the VQGAN and the MDC are optimized by:

$$\mathcal{L}(E, MDC) = \mathcal{L}_\text{rec} + \lambda_\text{det} \mathcal{L}_\text{det} + \lambda_\text{code} \mathcal{L}_\text{code} + \lambda_\text{per} \mathcal{L}_\text{per} + \lambda_\text{adv} \mathcal{L}_\text{adv}, \quad (9)$$

where $\mathcal{L}_\text{det} = ||\hat{\mathbf{S}}_{0.5} - \mathbf{S}_{0.5}||_1 + ||\hat{\mathbf{S}}_1 - \mathbf{S}_1||_1 + ||\hat{\mathbf{S}}_3 - \mathbf{S}_3||_1$. $\hat{\mathbf{S}}_\gamma$ and $\mathbf{S}_\gamma$ denote the estimated details and corresponding multi-luminance gradient by applying Gamma correction with $\gamma$ on the gradient. The reconstructed loss $\mathcal{L}_{rec}$ is based on $1 - \text{SSIM}(\hat{\mathbf{H}}, \mathbf{H})$ as it is effective as evidenced by [55]. $\mathcal{L}_\text{code} = ||\hat{z}_\mathbf{q} - z_\mathbf{q}||_2^2$, where $z_\mathbf{q}$ is the normal-light codebook features while $\hat{z}_\mathbf{q}$ is the reconstructed features.

**Shuffle Transformer Modulator.** The Shuffle Transformer Modulator (STM) aims to modulate features with different levels in the encoder of VQGAN. STM first exploits shuffle down [80] operation to re-scale different sizes to the smallest spatial size of the encoder of the VQGAN. Then, the shuffled features are concatenated and learned via $1 \times 1$ convolution followed by a Transformer. By inserting Transformer, STM is capable of modeling the global representation of the encoder of VQGAN to provide more useful features for the decoder of MDC (MDC-Dec) and GCA (GCA-Dec) for better enhancement. Figure 3(a) shows the STM.

**Deformable Transformer Perceptor with Encoder.** The Deformable Transformer Perceptor with Encoder (DTP-Enc) as shown

in Figure 3(b) explores the global content by inserting Transformers to effectively produce better offset, which is further perceived with the encoder features in VQGAN by deformable convolution to help improve the feature representation ability of the decoder of MDC/GCA. The DTP-Enc is conducted in each layer of different scales between the encoder and MDC-Dec/GCA-Dec (For simplicity, Figure 3 only draws the module at one scale).

## 4.3 Global Color Adjustment

To ensure the generated results with natural colors and better perceptual quality, we develop a simple yet effective Global Color Adjustment module (GCA), which is motivated by both low-light and normal-light images contain useful luminance and information that can help improve the colorfulness of results. Hence, we propose to build on spatially-varying representations between the estimated images in the GCA and low-light input ones to control the generated images for better perceptual quality and natural colors.

The GCA has a similar structure to MDC except for the input of the GCA decoder (GCA-Dec) and final outputs. For the input of GCA-Dec, with well-learned restored features after training MDC, we propose to deformable indicator by representative light to exploit the features $z_q$ to better guide the learning of the decoder of GCA. The GCA outputs exploitable adjustment image $Y_{GCA} \in \mathbb{R}^{H \times W \times 3}$ and corresponding weight map $M_{GCA} \in \mathbb{R}^{H \times W \times 1}$:

$$\{M_{GCA}, Y_{GCA}\} = GCA(F_E, z_q). \tag{10}$$

The learned $Y_{GCA}$ and $M_{GCA}$ are linearly added to the results in (8) to facilitate adjusting the generated results:

$$\hat{H} = \hat{H}_{vqgan} + \underbrace{M_{0.5} \cdot S_{0.5} + M_1 \cdot S_1 + M_3 \cdot S_3}_{\text{multi-luminance detail compensation}} + \underbrace{\alpha M_{GCA} Y_{GCA} + (1-\alpha) M_{GCA} L}_{\text{global color adjustment}},$$

$$\tag{11}$$

where $\hat{H}_{vqgan}$ can be served as the results of $\mathcal{N}_{content}(L)$ in (2); $M_{0.5} \cdot S_{0.5} + M_1 \cdot S_1 + M_3 \cdot S_3$ can be served as the results of $\mathcal{N}_{detail}(L)$ in (2); $\alpha M_{GCA} Y_{GCA} + (1-\alpha) M_{GCA} L$ can be served as the results of $\mathcal{N}_{color}(L)$ in (2). $\alpha \in [0, 1]$ is the global spatially-varying color adjustment factor to adjust the colors of enhanced results.

At this stage, parameters in the decoder of the VQGAN and the GCA are optimized by:

$$\mathcal{L}(D, GCA) = \mathcal{L}_{rec} + \lambda_{gca}\mathcal{L}_{gca} + \lambda_{per}\mathcal{L}_{per} + \lambda_{adv}\mathcal{L}_{adv}, \tag{12}$$

where $\mathcal{L}_{gca} = 1 - \text{SSIM}(Y_{GCA}, H)$.

**Deformable Indicator by Representative Light.** We propose the Deformable Indicator with Representative Light (DIRL) to guide the learning of GCA as the codebook has been optimized and the encoder can produce more representative features of normal light from low-light inputs after the stage of training the encoder and MDC. Thus the encoder features matched with the codebook via (3) can better represent the latent light of normal light, which will help the GCA learn better. The DIRL is shown in Figure 3(c).

**Triple Feature Transformation Module.** To better exploit useful features in decoders of MDC and GCA to improve the feature representation of the decoder in VQGAN for better image restoration, we develop the Triple Feature Transformation Module (TriFTM). The TriFTM is motivated by [62] and can effectively fuse useful features from MDC ($F_{MDC}$) and GCA ($F_{GCA}$) to the decoder of VQGAN, as shown in Figure 3(d). Notice that the TriFTM is only used in the

---

**Algorithm 1** Training Process of Our **PercepLIE**

// **Stage 1: Training VQGAN for LLR (Section 4.1)**
**Prepare:** Normal-light images **H**.
  **While** iter $\leq$ iter$_{max}^{vqgan}$ **do**:
      Encode normal-light images to latent **Codebook** via (3)
      Decode the learned **Codebook** to images via (4)
      Optimize**Encoder**, **Decoder**, and **Codebook** via (5)
      iter = iter +1
  **End while**
// **Stage 2: Tuning VQGAN Encoder with MDC (Section 4.2)**
**Prepare:** Low-light images **L**, normal-light images **H**, and learned **Codebook** and **Decoder** in **Stage1**.
  **While** iter $\leq$ iter$_{max}^{MDC}$ **do**:
      Generate multiple details and corresponding maps via (7)
      Compensate for details via (8)
      Optimize **Encoder** and **MDC** via (9)
      iter = iter + 1
  **End while**
// **Stage 3: Tuning VQGAN Decoder with GCA (Section 4.3)**
**Prepare:** Low-light images **L**, normal-light images **H**, and learned **Codebook** in **Stage 1**, and learned **Encoder** and **MDC** in **Stage 2**.
  **While** iter $\leq$ iter$_{max}^{GCA}$ **do**:
      Generate exploitable adjustment images and correspond map via (10)
      Optimize **GCA** and **Decoder** via (11)
      iter = iter + 1
  **End while**

---

stage of training GCA, while the stages of training VQGAN and MDC use ResBlocks [11] in the decoder of VQGAN.

## 5 Experiments

In this section, we compare our PercepLIE with 17 state-of-the-art (SOTA) approaches. Extensive analysis is also conducted to verify the effectiveness of our PercepLIE.

### 5.1 Datasets

**Synthetic datasets.** LOL dataset [66] is a widely used synthetic dataset with 485/15 training/testing samples. We use it to evaluate the enhancement performance on the synthetic scenes. Following [63], we also use VE-LOL dataset [38] to examine the models' generalization on unseen scenes by using the model trained on LOL. VE-LOL contains two sub-datasets: VE-LOL-real and VE-LOL-synthetic, which respectively contain 400/100 and 900/100 training/testing samples. Such large data (a total of 1,500 pairs of images) is enough to verify the models' generalization.

**Real-world datasets.** DICM [33], LIME [10], MEF [42], and NPE [61] are widely used real-world datasets. We use them to evaluate the effectiveness of the proposed method.

### 5.2 Training Details

**Stage 1 for VQGAN (Section 4.1):** We train the VQGAN with 2,000K iterations, i.e., iter$_{max}^{vqgan}$=2,000K, on LOL normal-light samples [66], where data augment (e.g., flip, randomly crop, and rotate)

**Table 1: Comparisons with SOTAs on synthetic datasets. The main metrics NIQE, PI, and LPIPS, evaluating naturalness and perceptual quality, are presented. PSNR and SSIM are also provided for reference. The best results are marked in red. ↑ (↓) means higher (lower) is better. Adj. denotes whether the method is adjustable or not. Cross-dataset verification means the model trained on LOL is used to verify models' generalization in unknown scenes. All the metrics are ensured to be re-computed using the same codes for fair comparisons.**

(a) Comparisons with SOTA methods on the LOL [66].

| Methods | Venues | Adj. | NIQE ↓ | PI ↓ | LPIPS ↓ | SSIM ↑ | PSNR ↑ |
|---|---|---|---|---|---|---|---|
| RetinexNet [66] | BMVC'18 | ✗ | 8.932 | 4.975 | 0.465 | 0.548 | 18.64 |
| KinD [82] | MM'19 | ✗ | 4.871 | 3.496 | 0.208 | 0.839 | 18.67 |
| DeepUPE [60] | CVPR'19 | ✗ | 7.523 | 4.508 | 0.367 | 0.531 | 13.16 |
| MIRNet [77] | ECCV'20 | ✗ | 4.864 | 3.867 | 0.233 | 0.837 | 22.28 |
| FIDE [71] | CVPR'20 | ✗ | 4.712 | 3.654 | 0.181 | 0.855 | 21.53 |
| ZeroDCE [9] | CVPR'20 | ✗ | 7.946 | 4.475 | 0.384 | 0.613 | 17.40 |
| KinD++ [81] | IJCV'21 | ✗ | 4.754 | 3.913 | 0.197 | 0.820 | 18.27 |
| Enlighten [22] | TIP'21 | ✗ | 4.952 | 3.912 | 0.310 | 0.721 | 19.57 |
| ZeroDCE++ [34] | TPAMI'21 | ✗ | 7.842 | 4.475 | 0.384 | 0.611 | 15.72 |
| RUAS [39] | CVPR'21 | ✗ | 6.257 | 4.481 | 0.256 | 0.671 | 17.05 |
| SwinIR [37] | ICCVW'21 | ✗ | 4.631 | 4.082 | 0.150 | 0.831 | 19.42 |
| SCL [36] | AAAI'22 | ✗ | 7.632 | 4.563 | 0.331 | 0.606 | 13.24 |
| UNIE [25] | ECCV'22 | ✗ | 4.598 | 4.298 | 0.139 | 0.827 | 22.92 |
| SCI [43] | CVPR'22 | ✗ | 7.872 | 4.544 | 0.333 | 0.621 | 15.40 |
| SNR [72] | CVPR'22 | ✗ | 5.177 | 4.283 | 0.150 | 0.896 | 27.09 |
| Restormer [76] | CVPR'22 | ✗ | 4.658 | 4.109 | 0.139 | 0.883 | 24.93 |
| LLFlow [63] | AAAI'22 | ✗ | 5.583 | 4.418 | 0.114 | 0.925 | 27.96 |
| **PercepLIE** | - | ✔ | **4.496** | **3.382** | **0.080** | 0.895 | 25.09 |

(b) Cross-dataset verification on the VE-LOL [38].

| Methods | Venues | Adj. | NIQE ↓ | PI ↓ | LPIPS ↓ | SSIM ↑ | PSNR ↑ |
|---|---|---|---|---|---|---|---|
| RetinexNet [66] | BMVC'18 | ✗ | 9.774 | 6.012 | 0.756 | 0.425 | 13.58 |
| KinD [82] | MM'19 | ✗ | 5.285 | 4.131 | 0.438 | 0.674 | 17.98 |
| DeepUPE [60] | CVPR'19 | ✗ | 9.572 | 5.859 | 0.597 | 0.483 | 14.50 |
| MIRNet [77] | ECCV'20 | ✗ | 5.554 | 4.017 | 0.509 | 0.611 | 18.16 |
| FIDE [71] | CVPR'20 | ✗ | 6.154 | 4.293 | 0.467 | 0.653 | 17.80 |
| ZeroDCE [9] | CVPR'20 | ✗ | 9.251 | 5.565 | 0.611 | 0.537 | 16.61 |
| KinD++ [81] | IJCV'21 | ✗ | 5.142 | 4.431 | 0.435 | 0.670 | 16.86 |
| Enlighten [22] | TIP'21 | ✗ | 6.668 | 4.590 | 0.567 | 0.574 | 17.04 |
| ZeroDCE++ [34] | TPAMI'21 | ✗ | 9.071 | 5.496 | 0.612 | 0.542 | 17.13 |
| RUAS [39] | CVPR'21 | ✗ | 6.143 | 5.982 | 0.608 | 0.472 | 13.25 |
| SwinIR [37] | ICCVW'21 | ✗ | 6.409 | 4.564 | 0.511 | 0.613 | 16.54 |
| SCL [36] | AAAI'22 | ✗ | 9.163 | 5.609 | 0.613 | 0.516 | 14.86 |
| UNIE [25] | ECCV'22 | ✗ | 5.162 | 4.001 | 0.437 | 0.692 | 18.40 |
| SCI [43] | CVPR'22 | ✗ | 9.227 | 5.698 | 0.624 | 0.494 | 15.34 |
| SNR [72] | CVPR'22 | ✗ | 5.097 | 4.235 | 0.454 | 0.686 | 21.54 |
| Restormer [76] | CVPR'22 | ✗ | 5.882 | 4.738 | 0.458 | 0.665 | 20.35 |
| LLFlow [63] | AAAI'22 | ✗ | 6.358 | 4.462 | 0.459 | 0.658 | 19.76 |
| **PercepLIE** | - | ✔ | **4.854** | **3.942** | **0.422** | 0.671 | 21.67 |

| (a) Input | (b) GT | (c) SCL [36] | (d) SNR [72] | (e) SCI [43] | (f) UNIE [25] | (g) LLFlow [63] | (h) **PercepLIE** |

**Figure 4: Visual comparisons with recent SOTAs on LOL [66] and VE-LOL [38] datasets. From top to bottom: LOL, VE-LOL-real, and VE-LOL-synthetic. Our PercepLIE is able to generate much clearer images with finer details while existing methods either hand down noises or lose details or generate under- or over-results. Best viewed on high-resolution display with zoom-in.**

is used. The number of codebooks, i.e., $N$, is 1024, and the dimension of the codebook, i.e., $n_z$, is 512. The learning rate is $1 \times 10^{-4}$. $\lambda_{per}$ and $\lambda_{adv}$ are set as 1 and 0.1. The batch size is 8.

**Stage 2 for MDC (Section 4.2):** At the training MDC stage, the model is trained for 120K iterations, i.e., $\text{iter}_{max}^{MDC}$=120K, on the LOL dataset. $\lambda_{det}$ and $\lambda_{code}$ are set as 0.1 and 1, respectively. The $\gamma$ in Gamma correction is set as 0.5, 1, and 3 to derive details with different luminance. The batch size is 4.

**Stage 3 for GCA (Section 4.3):** At the training GDA stage, the model is also trained for 120K iterations, $\text{iter}_{max}^{GCA}$=120K, on the LOL dataset. $\lambda_{gca}$ is set as 0.1. The batch size is 2. The $\alpha$ is set as 0.8

when training. For inference, unless otherwise stated, $\alpha = 0.8$ is the default. Section 5.6 analyzes the effectiveness and flexibility of $\alpha$.

We use Swin Transformer block [40] with depth 2 as our Transformer block. The patch size is $256 \times 256$ for all the experiments. All the experiments are optimized via ADAM [30] and trained on two NVIDIA 3090 GPUs based on PyTorch [45]. Algorithm 1 describes the training process of our PercepLIE.

## 5.3 Results on Synthetic Datasets

Table 1 summarises the comparison results with SOTA methods on LOL [66] and cross-dataset verification results on the VE-LOL [38].

**Table 2: Comparisons with recent SOTAs on real-world datasets in terms of NIQE and PI. PercepLIE ($\alpha$) means setting $\alpha$ in (11) at inference. As our PercepLIE is adjustable, adjusting $\alpha$ can generate results with different perceptual qualities. Note that the model with $\alpha = 0$ performs the best, which indicates that low-light images can provide useful content to improve results toward better colors and perceptual quality, further suggesting the superiority of our PercepLIE.**

| Methods | SwinIR [37] | Zero-DCE++ [34] | RUAS [39] | Restormer [76] | SCL [36] | SNR [72] | SCI [43] | UNIE [25] | LLFlow [63] | PercepLIE (0) | PercepLIE (0.8) | PercepLIE (1) |
|---|---|---|---|---|---|---|---|---|---|---|---|---|
| NIQE ↓ | 4.46 | 4.49 | 6.27 | 4.58 | 4.23 | 4.64 | 4.15 | 4.85 | 4.23 | **4.11** | 4.24 | 4.31 |
| PI ↓ | 3.56 | 3.36 | 4.76 | 4.40 | 3.21 | 3.84 | 3.17 | 4.09 | 3.26 | **3.07** | 3.18 | 3.26 |

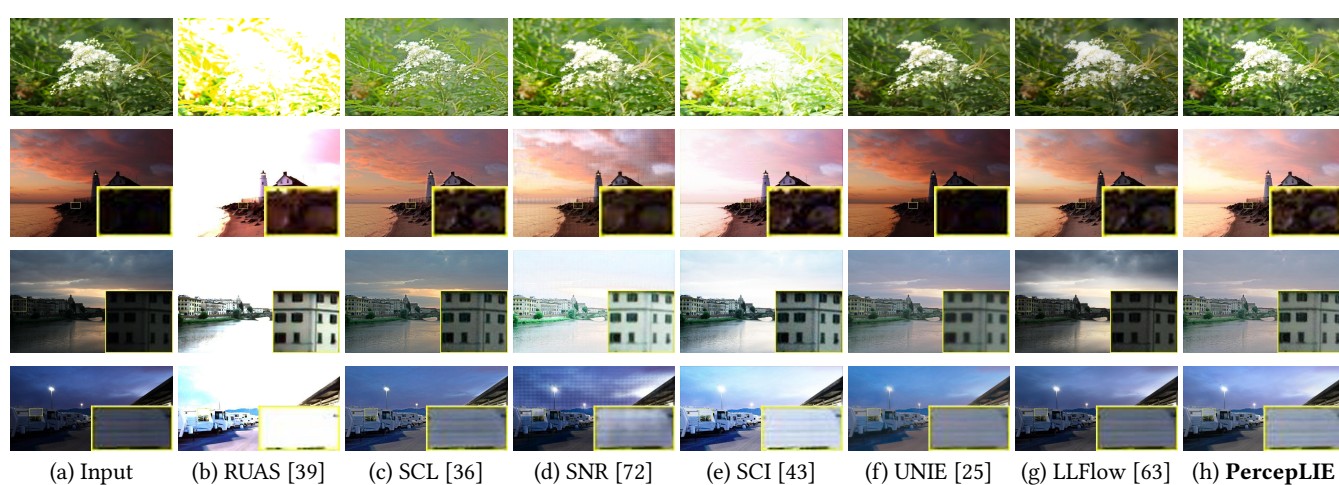

| (a) Input | (b) RUAS [39] | (c) SCL [36] | (d) SNR [72] | (e) SCI [43] | (f) UNIE [25] | (g) LLFlow [63] | (h) **PercepLIE** |
|---|---|---|---|---|---|---|---|

**Figure 5: Visual comparisons with recent SOTAs on real-world datasets. Our PercepLIE is capable of generating clearer results with finer details . Best viewed on high-resolution display with zoom-in.**

Both Table 1(a) and Table 1(b) show that our PercepLIE achieves the best NIQE, PI, and LPIPS, the metrics of measuring naturalness and perceptual quality, indicating the proposed algorithm is a better LIE model and has the best generalization with better naturalness and perceptual quality to unknown scenes. Visual results in Figure 4 show that our PercepLIE is capable of restoring results with better naturalness and preserving finer details while existing approaches always either hand down extensive noise or produce under-enhanced results or lose image details.

## 5.4 Results on Real-World Datasets

Table 2 reports the comparison results on real-world datasets in terms of average NIQE and PI. Notice although the default method with $\alpha = 0.8$ is not the best but is adjustable, we can adjust $\alpha$ to generate satisfactory results. We note that the PercepLIE with $\alpha = 0$ reaches the best, which indicates that low-light images can provide useful content to improve results toward better naturalness and perceptual quality, further suggesting the superiority and flexibility of the PercepLIE. In Figure 5, these visual examples illustrate that our PercepLIE always produces more natural and realistic quality with finer structures while existing algorithms always generate kinds of unsatisfactory results.

## 5.5 Ablation Study

**Effectiveness of MDC and GCA.** As MDC and GCA modules are our proposed main contributions, it is of great interest to analyze their effect. Table 3 reveals that the model without both MDC and GCA (Table 3**(a)**) produces worse results and the model without

**Table 3: Effect on Multi-luminance Detail Compensation (MDC) and Global Color Adjustment (GCA).**

| Experiment | NIQE ↓ | PI ↓ | LPIPS ↓ | SSIM ↑ | PSNR ↑ |
|---|---|---|---|---|---|
| **(a)** w/o **MDC** & w/o **GCA** | 4.886 | 4.161 | 0.137 | 0.849 | 23.07 |
| **(b)** w/o **MDC** & w/ **GCA** | 4.807 | 4.059 | 0.114 | 0.856 | 23.54 |
| **(c)** w/o $\gamma = 0.5$ in **MDC** & w/ **GCA** | 4.663 | 3.881 | 0.108 | 0.887 | 24.61 |
| **(d)** w/o $\gamma = 1$ in **MDC** & w/ **GCA** | 4.713 | 3.857 | 0.094 | 0.882 | 24.14 |
| **(e)** w/o $\gamma = 3$ in **MDC** & w/ **GCA** | 4.686 | 3.608 | 0.102 | 0.874 | 24.42 |
| **(f)** w/ **MDC** & w/o **GCA** | 4.432 | 3.417 | 0.093 | 0.884 | 24.71 |
| **(g)** w/ **MDC** & w/ **GCA** (*Ours*) | 4.496 | 3.382 | 0.080 | 0.895 | 25.09 |

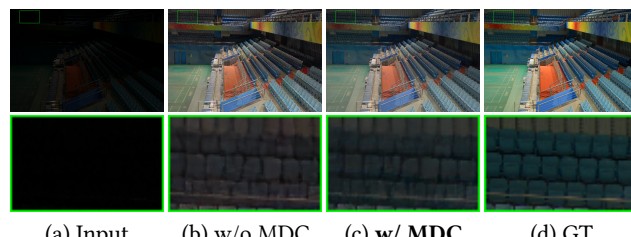

| (a) Input | (b) w/o MDC | (c) **w/ MDC** | (d) GT |
|---|---|---|---|

**Figure 6: With MDC, our model recovers results with finer details (c). Best viewed with zoom-in.**

either MDC (Table 3**(b)**) or GCA (Table 3**(f)**) is not more effective than full model (Table 3**(g)**). It is also notable that each luminance detail in MDC can boost the enhancement quality. These results illustrate that both MDC and GCA are important for the LIE task. In Figure 6, our full model is able to recover more natural results with finer details, while the model without MDC tends to lose structure.

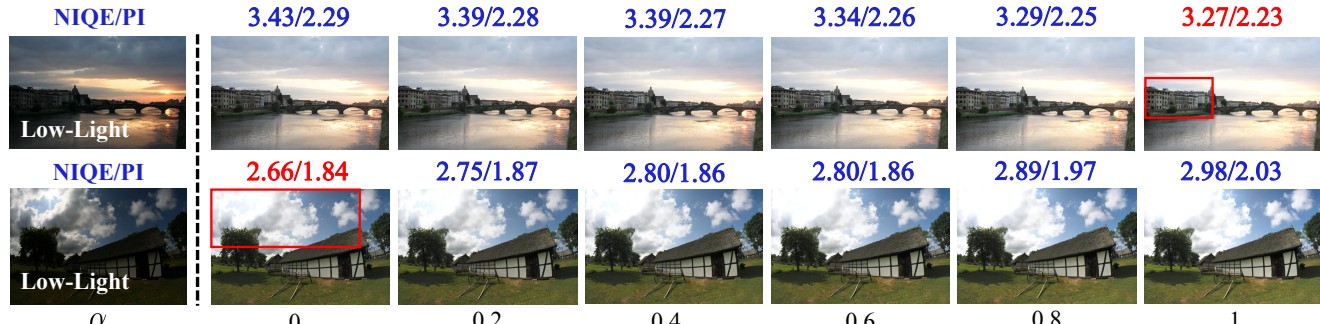

**Figure 7: Adjustable results on real-world scenarios. By adjusting the global spatially-varying color adjustment factor $\alpha$, our PercepLIE is able to generate results with more natural colors and better perceptual quality according to different low-light inputs with different illuminations. Best viewed on high-resolution display with zoom-in.**

**Table 4: Effect on DTP-Enc, DIRL, and TriFTM.**

| Experiment | NIQE ↓ | PI ↓ | LPIPS ↓ | SSIM ↑ | PSNR ↑ |
|---|---|---|---|---|---|
| (a) Disabling DTP-Enc | 4.901 | 3.543 | 0.142 | 0.851 | 22.98 |
| (b) Disabling DIRL | 4.613 | 3.317 | 0.097 | 0.883 | 24.73 |
| (c) Disabling TriFTM | 4.772 | 3.754 | 0.094 | 0.881 | 23.98 |
| (d) Full model (*Ours*) | 4.496 | 3.382 | 0.080 | 0.895 | 25.09 |

**Effectiveness of deformable Transformer preceptor with encoder (DTP-Enc).** DTP-Enc is used to help the decoder of MDC and DCA learn more useful features from the encoder of VQGAN for better enhancement. Table 4 shows that our DTF-Enc performs better when we disable the module (Table 4**(a)** vs. **(d)**).

**Effectiveness of deformable indicator by representative light (DIRL).** The DIRL aids the GCA at **Stage 3** by utilizing the codebook from normal-light images to furnish superior light information. The encoder effectively restores normal-light features from low-light inputs after completing the training phase with the encoder and MDC. Subsequently, the restored features matched with the codebook via (3) can more appropriately represent latent features of normal light, resulting in better learning for the GCA. Table 4 illustrates that our DIRL is effective for better image enhancement (Table 4**(b)** vs. **(d)**).

**Effectiveness of triple feature transformation module (TriFTM).** In **Stage 3**, the TriFTM is employed to ameliorate the decoder's representation capability in VQGAN. This is accomplished by acquiring relevant features from MDC-Dec and GCA-Dec decoders which will help improve the overall effectiveness of the decoder in the VQGAN. Table 4 shows that disabling the TriFTM decreases the enhancement performance (Table 4**(c)** vs. **(d)**).

### 5.6 Adjustability

As our PercepLIE uses the global spatially-varying color adjustment factor $\alpha$ in GCA to adjust the restoration results for better perceptional quality and natural colors, it is necessary to examine its effectiveness and flexibility. Figure 8 shows the NIQE and PI results with different $\alpha$. Note that it is difficult to find an $\alpha$ that works best for NIQE, PI, and LPIPS simultaneously. We argue that our PercepLIE is flexible as we can dynamically adjust $\alpha$ to generate visually-pleasing results according to different low-light inputs. Figure 7 shows two real-world examples, where we adjust the $\alpha$

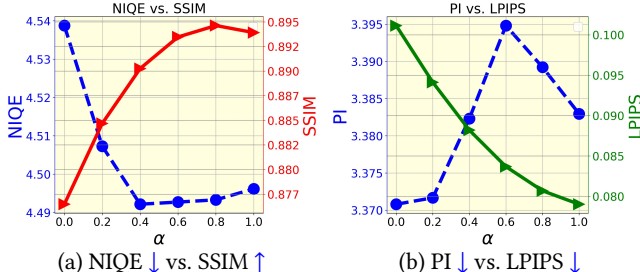

**Figure 8: Adjustability on the LOL [66]. We adjust $\alpha$ to generate results with different perceptual qualities.**

to generate results with different colors and perceptual quality. By adjusting $\alpha$, the PercepLIE is able to produce better results with better colors and naturalness according to different low-light inputs with different illumination. In Figure 7, higher $\alpha$ for the first example results in better performance, while lower $\alpha$ generates superior results for the second example. The two examples clearly suggest that our PercepLIE is flexible and superior when handling different low-light inputs with different illuminations.

## 6 Conclusion

We have presented a new path to perceptual LIE, called **PercepLIE**, by introducing the VQGAN with multi-luminance detail compensation (MDC) and global color adjustment (GCA). The VQGAN is used to explore the latent light representations of normal-light images to guide the deep models for better image enhancement. The MDC generates diverse details to facilitate accurate structural detail estimation, while the GCA is employed to ensure natural colors and better perceptual quality in the generated images. Experiments demonstrate that our PercepLIE outperforms existing techniques on synthetic and real-world datasets based on various perceptual quality metrics, including NIQE, PI, and LPIPS.

## Acknowledgements

This work was supported by the National Natural Science Foundation of China No. 62306343, Shenzhen Science and Technology Program (No. KQTD20221101093559018), and Fundamental Research Funds for the Central Universities, Sun Yat-sen University under Grants No. 23qnpy56.

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
