# OpenReview forum: "PercepLIE: A New Path to Perceptual Low-Light Image Enhancement"
_acmmm.org/ACMMM/2024/Conference — MM2024 Poster_

### Official Review · Reviewer_a8E5 · 2024-05-05

**Rating:** 4
**Confidence:** 3

**Summary:**

This paper proposes a perceptual low-light image enhancement by constructing a network to address (1) multiple gamma adjusted details and (2) global color adjustment. Two networks (MDC and GCA) are proposed to address the details part and the color parts based on the latent light representation (VQGAN). Experiments show the proposed method outperforms several other methods in several performance metrics.

**Strengths:**

This paper proposes to utilize the different revealed details via different level of gamma correction to improve the quality. The color is also addressed. The proposed framework works on the latent light representation, which is trained on low light database.

**Limitations:**

(1) This proposed result shows best performance in several considered performance metrics, but not all.  Those numbers are objective evaluation. It will be good to have real human subjects to evaluate the performance. From Figure 4, PerceptLIE seems having large color distortion on sky area comparing to GT.   Figure 5, the PerceptLIE seems color de-saturated with dark lift comparing to other methods. The effectiveness on human subjective testing remain unclear.
(2) How to select the set of gamma correction parameters? The paper shows using 0.5, 1.0, and 3.0.  Any particular reason/experiments to show these 3 are the best?  And for applying gamma correction process, does the algorithm needs to de-gramma back to linear domain, and apply gamma correction again, or just apply gamma correction on the existing picture?
(3) In table 3, the authors compare with and without VQ.  Please elaborate how to select n_z (feature dimension) so the performance can be optimized.

**Suitability:**

2

---

### Official Review · Reviewer_2z1f · 2024-05-13

**Rating:** 5
**Confidence:** 3

**Summary:**

This paper proposes a perceptual low-light image enhancement algorithm. This algorithm observes an interesting and natural phenomenon that the latent light codebook representation of low-light images is quite different from those of images captured in normal light. Hence, enforcing normal-light codebook priors to guide the deep models will help better enhance. To that end, the authors present the VQGAN with Multi-luminance Detail Compensation (MDC) and Global Color Adjustment (GCA). By combining the VQGAN with MDC and GCA within a stage-wise training mechanism, our method generates images with finer details and natural colors and achieves favorable performance on both synthetic and real-world datasets in terms of perceptual quality metrics including NIQE, PI, and LPIPS.

**Strengths:**

1, This paper is well written an easy to follow and the organization is clear.

2, The observed codebook prior in the low and normal light is interesting and meaningful, which is an inherent property for low-light image enhancement.

3, The results look better. Although current existing algorithms are driven by pursuing higher PSNR and SSIM, these solutions are usually not effective to handle cross-domain images, e.g., training on one synthetic dataset and then test on another synthetic testing date or training on synthetic dataset and then test on real images. The paper is effective on the above problems.

4, Perceptual quality obtained by this method is better and the method is better flexible.

**Limitations:**

1, How the authors to adjust the controllable results? The authors should specify this more detailedly.

2, The network structure looks complicated. To eliminate the confusion of different modules, the authors should provide more explanation on each component.

3, How do the authors obtain the codebook priors? And how do the author utilize the priors? It seems to be not specific enough.

**Suitability:**

2

---

### Official Review · Reviewer_hjwU · 2024-05-21

**Rating:** 6
**Confidence:** 4

**Summary:**

This paper proposes a new path to perceptual low-light Image enhancement, which explores the VQGAN to low-light image enhancement task. To better generate perceptual image quality, the authors propose Multi-luminance Detail Compensation (MDC) and Global Color Adjustment (GCA). The MDC module is used to facilitate better detail estimation while the GCA module is used to optimize the colors of low-light input images. The experiments on both synthetic and real-world datasets demonstrate that the proposed PercepLIE is able to produce better perceptual image quality in terms of perceptual quality metrics including NIQE, PI, and LPIPS.

**Strengths:**

1, This paper is well-motivated. The observation that latent light features of the low-light images are quite different from those captured in normal light is meaningful and natural property of low-light images and normal-light images.

2, This paper is well-written and this paper is easy to follow. The writing is professional and the logic of this paper is easy to understand.

3, The experiments are adequate and the results are good.

4, This network design is interesting.

**Limitations:**

1, How do the authors adjust the result? I am confused about how to control the results.

2, How do authors get the codebook of low-light images and normal-light images?

3, Will the source codes be released?

4, The authors should conclude the limitations of this paper.

**Suitability:**

3

---

### Meta-Review · Area_Chair_61xv · 2024-06-30

**Recommendation:** Accept (Poster)
**Confidence:** 5

**Metareview:**

All reviewers suggest to accept the paper.